# SoundSynp: Sound Source Detection from Raw Waveforms with Multi-Scale Synperiodic Filterbanks

## Abstract

Accurately estimating sound sources' temporal location, spatial location and semantic identity label from multi-channel sound raw waveforms is crucial for an agent to understand the 3D environment acoustically. Multiple sounds form a complex waveform mixture in time, frequency and space, so accurately detecting them requires a representation that can achieve high resolutions across all these dimensions. Existing methods often cannot do it well because they either extract hand-engineered features (*i.e.* STFT, LogMel) that require a great deal of parameter tuning work, or propose to learn a single-scale filter bank to process sound waveforms that has limited time-frequency resolution capability. In this paper, we tackle this issue by proposing to learn a group of parameterized synperiodic filter banks. Each synperiodic filter's length and frequency resolution are inversely related, hence is capable of maintaining a better time-frequency resolution trade-off. By alternating the periodicity term, we can easily obtain a group of synperiodic filter banks, where each bank differs in its temporal length. Convolution of the proposed filterbanks with the raw waveform helps to achieve multi-scale perception in the time domain. Moreover, applying synperiodic filter bank to recursively process a downsampled waveform enables to achieve multi-scale perception in the frequency domain. Benefiting from the multi-scale perception in both time and frequency domain advantage, our proposed synperiodic filter bank group learns a dynamic multi-scale time-frequency representation in a data-driven way. Following synperiodic filter bank group front-end, we add a Transformer-like backbone with two parallel soft-stitched branches to learn semantic identity label and spatial location representation semi-independently. Experiments on both direction of arrival estimation task and the physical location estimation task shows our framework outperforms existing methods by a large margin. Replacing existing methods' front-end with synperiodic filter bank also helps to improve the performance.

## 1 Introduction

The fundamental task for an agent to perceive and interact with the 3D environment is to know the location and semantic identity of its nearby objects. The location includes spatial location that is either stationary or moving, temporal location like start time and end time. Vision-based such environment perception has received large attention in the past decade and we have witnessed huge progress in tasks such as object detection(Liu et al., 2016; Lin et al., 2014; Yang et al., 2019), classification(He et al., 2016a) and tracking(Wang et al., 2019). Nevertheless, the sound-based counterpart research has far lagged behind, despite all the fascinating properties sound signal exhibits. For example, sound is ubiquitous and insensitive to ambient illumination change, it has no field-of-view (FoV) constraints and is capable of circumventing physical barriers to perceive scene beyond line-of-sight. As a complementary sensing approach to vision, sound-based environment perception is of vital importance for acoustic scene understanding. A typical example is the sound source detection (answer where is it, when does it happen and what is it), given the recorded multi-channel sound waveforms.

To detect sound sources, we often deploy a spatially-configured microphone array to record an acoustic environment. Unlike camera or LiDAR scanner that directly captures RGB image or measures the range distance, sound waveform itself is a highly compressed one-dimensional points

with high sampling rate, all sound source signals are compressed and mixed into the one-dimensional data format. Since different sound sources have different frequency property, it is essential convert waveform into time-frequency representation so that frequencies hidden in the waveform are explicitly split out. This is often achieved by projecting the raw waveform into various orthogonal frequency basis. At the same time, a sound source's spatial location clue lies in inter-channel difference of recorded waveforms (*i.e.* phase difference in this work). It is essential to design a neural network that jointly encodes mono-channel time-frequency representation and inter-channel phase difference from the raw waveforms in a unified, parameter-frugal and computation-efficient manner. The learned representation should have elegant resolution in both time, frequency and space domain so that sound sources can be precisely detected.

However, learning such representation is a tough task. Challenges derive from both theoretical side and practical side. According to Uncertainty Principle, we cannot achieve the optimal resolution in time and frequency domain at the same time, but instead keep a trade-off between them. Traditional hand-engineered sound feature(Davis & Mermelstein, 1980; Cao et al., 2021; Tho Nguyen et al., 2020; Brandstein & Silverman, 1997) and some recently proposed learnable filter bank(Ravanelli & Bengio, 2018; Zeghidour et al., 2018) empirically set the same length for all filters, resulting in human-biased, unadjustable time-frequency resolution map. Some other work(Zeghidour et al., 2021) correlates filter frequency response and filter length by initializing in mel-scale, but it is not scalable nor stable because the final initialization depends on the filter number. Moreover, all existing methods process raw waveform with one-scale filter bank, we argue that the one-scale sound perception easily leads to incomprehensive sound sources sensing, especially when sound sources have different frequency property or undergo various spatial motion.

In this paper, we first give comprehensive theoretical analysis on the filter bank impact on its extracted feature's time and frequency resolution. Based on the analysis, we propose a simple yet effective synperiodic filter bank construction strategy in which each filter's frequency response and length are correlated by rotating periodicity such that each filter's length is inversely proportional to its frequency resolution. The synperiodic filter bank thus internally maintains a better time-frequency resolution trade-off than traditional fixed-length filter bank. Coupling the filter length with its frequency response helps us to reduce human intervention in filter bank design. By simply alternating the periodicity term, we further construct a group of synperiodic filter banks, with which we achieve multi-scale perception in time domain. At the same time, by applying a synperiodic filter bank to process one raw waveform as well as its consecutively-downsampled versions, we achieve multi-scale perception in frequency domain. The multi-scale perception in both time and frequency domain of synperiodic filter bank enables the neural network to dynamically learn better representation for sound source detection in a data-driven way. It is worth noting that synperiodic filter bank parameter number is just linear to filter number (adds up to less than $1\%$ of the whole parameters) and it can be efficiently implemented as a 1D convolution operator.

Following the aforementioned learnable front-end, we add a Transformer-like backbone network with two paralleling branches with intermediate soft-parameter sharing to learn sound source's semantic and spatial location related representation both jointly and separately. Experiment on both direction-of-arrival (DoA) task and physical location estimation task shows that our proposed framework outperforms comparing methods by a large margin. Replacing existing method's head with our proposed synperiodic filter bank also improvdes the performance.

## 2 RELATED WORK

Sound signal processing has been thoroughly studied in traditional digital signal processing area. The preliminary step of sound signal processing is usually to convert raw waveform into 2D time-frequency representation. There are two main realms: Fourier transform based and Wavelet based transform(Sturm, 2007). Traditional sound feature design are motivated influenced by human-auditory system. For example, they often convert frequency bins into mel-scale to imitate human hearing system, like MFCC(Davis & Mermelstein, 1980), LogMel(Cao et al., 2021; Grondin et al., 2019), the filter length is empirically chosen and often a windowing is added to avoid spectrum leakage. For inter-channel phase difference encoding, it is often recommended to encode in frequency domain due to the less-computation advantage. Typical phase difference features include GCC-Phat(Brandstein & Silverman, 1997) and intensity vector(Cao et al., 2021).

Sound source detection has been previously treated as sound structure(Thrun, 2006) estimation, sound object detection(He et al., 2021) and sound event detection and localization (SELD) problem(Adavanne et al., 2018; Thi Ngoc et al., 2021). It involves jointly identifying a sound source's semantic label and predicting its spatial location, the two sub-tasks have been thoroughly studied separately in acoustics(Nandwana & Hasan, 2016; Mohan et al., 2008; Sundar et al., 2020; Vera-Diaz et al., 2018) and computer vision community(He et al., 2016a). T. Kim *et al.*(Kim et al., 2019) provided a review and discussion for raw-audio based event classification. Benefiting from the success of the traditional hand-engineered sound feature and the large availability of mature image-based deep neural networks(He et al., 2016a), most work(Grondin et al., 2019; Cao et al., 2021; Grondin et al., 2019; Adavanne et al., 2018; Thi Ngoc et al., 2021) tackle the task by first extracting hand-engineered sound feature and then feeding them to mature image-based neural networks. This workflow is straightforward and often guarantees reasonably good results but it is not end-to-end trainable and heavily depend on image side which may not be optimal for sound processing. At the same time, some work(He et al., 2021; Adavanne et al., 2018; Tho Nguyen et al., 2020) have simplified the problem by assuming no two sound sources of the same semantic label but different spatial location happen at the same time. This assumption avoids semantic label and spatial location association issue but may not reflect real scenarios.

In recent years, a bunch of work tried to design neural work to directly learn from raw sound waveform, ranging from the earlier methods that directly apply stacked layers to process raw waveform(Schneider et al., 2019; Palaz et al., 2013; Jaitly & Hinton, 2011; Sainath et al., 2013) to the recent frequency-sensitive filter bank learning methods(Zeghidour et al., 2021; Ravanelli & Bengio, 2018; Zeghidour et al., 2018; He et al., 2021; Hoshen et al., 2015; Sainath et al., 2015; Luo & Mesgarani, 2019). The filter bank parameter is initialized in either mel-scale(Ravanelli & Bengio, 2018; Zeghidour et al., 2021; He et al., 2021; Zeghidour et al., 2018) or as Gammatone filter(Hoshen et al., 2015; Sainath et al., 2015). SoundDet(He et al., 2021) is the first work to directly learn from multi-channel raw waveforms to detect sound sources. It designs MaxCorr filter bank to directly convolve with multi-channel raw waveforms to learn phase difference aware features.

Multi-scale representation has a rich history in computer vision community ((Liu et al., 2016; 2021; Lin et al., 2016; He et al., 2016b)), in which multi-scale representation strategy has been proposed to accommodate large object scale variation. For example, SSD(Liu et al., 2016) combines feature maps of various scale to cover potential objects of various sizes. In addition to spatial multi-scale representation, feature space multi-scale representation has been explored as well. For example, He *et. al.*(He et al., 2016b) incorporate features arising from intermediate layers to boost attribute recognition. In sound signal processing, Stéphane Mallat(Bruna & Mallat, 2013; Mallat, 2012) proposed wavelet scattering to obtain multi-scale sound representation by iteratively treating the proceeding processed sound waveform as new virtual waveform for further process.

## 3 SOUND SOURCE DETECTION FROM MULTI-CHANNEL RAW WAVEFORMS

### 3.1 SOUND SOURCE DETECTION PROBLEM DEFINITION

We use a spatially-deployed microphone array, like four closely-bounded microphones in a circular coplanar configuration, to record an acoustic environment where various sound sources undergo independent and unconstrained spatial motion. Recorded multi-channel raw waveforms are represented as $W = \{x_i(t)\}_{i=1}^4$, each waveform is 1-d vector of size $T$ sampled at a fixed sampling rate. We denote $K$ sound sources by $S = \{s_k = (t_s^k, t_e^k, sp_k, sl_k)\}_{k=1}^K$, one single sound source corresponds to a start time $t_s$, end time $t_e$, stationary or moving spatial motion $sp$ and a semantic identity label $sl$. Our goal is to learn a representation $\mathcal{P}$ that is representative enough to detect sound sources both spatially and semantically. $\mathcal{P}$ is a temporal framewise representation of shape $T_1 \times N$. We propose to learn $\mathcal{P}$ directly from the raw waveforms via a deep neural network $\mathcal{F}$ parameterized by $\theta$,

$$\mathcal{P}_{T_1 \times N} = \mathcal{F}(W_{T \times 4} | \theta), \quad T_1 \ll T \tag{1}$$

A learned elegant representation should have large resolution in both time, frequency and space domain and exhibit strong capability at addressing three intrinsic challenges: Polyphonicity, sound sources are mutually independent, they can freely overlap temporally; Time-scale variance, the

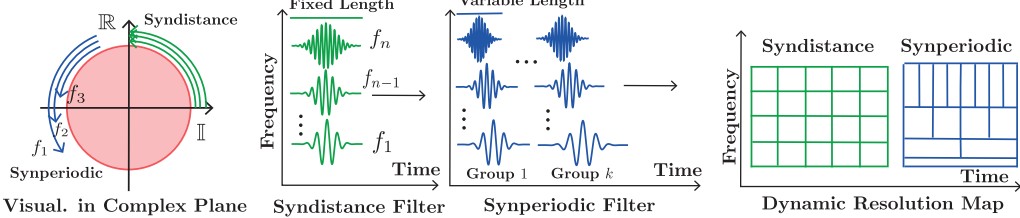

Figure 1: Synperiodic filter bank illustration: Syndistance filter bank (green color) rotates the same distance in complex plane and thus has the same kernel length, regardless of frequency it carries. Its time-frequency dynamic resolution map is thus rectangular. our proposed synperiodic filter bank (blue color) are generated by rotating the same periodicity number. So filter carrying lower frequency has larger kernel size than those with higher frequency response. As a result, synperiodic filter bank's time-frequency dynamic resolution map that is able to achieve a better trade-off than traditional syndistance filter bank.

temporal duration of different sound sources varies enormously, from seconds to minutes long; Datasize, sound waveform is sampled with large sampling rate, resulting in large data size.

Sound source presence clue lies in mono-channel time-frequency representation and its spatial motion clue lies in inter-channel interaction, like phase difference. Our proposed synperiodic filter bank group jointly encodes time-frequency representation and phase difference from raw waveforms in a unified way. More detail about phase difference encoding is given in the Appendix A.1.

## 3.2 MULTI-SCALE SYNPERIODIC FILTER BANK

We denote the general filter bank of $M$ filters by $\mathcal{F} = \{\mathcal{F}_i(w_i, \sigma_i, l_i) = \phi(w_i) \cdot \omega(\sigma_i)\}_{i=1}^M$. Each single filter $\mathcal{F}_i$ is created by multiplying the frequency-selective filter $\phi(w_i)$ of frequency response $w_i$ (*i.e.* sinusoidal basis or band-pass basis(Ravanelli & Bengio, 2018)) with a windowing function $\omega(\sigma_i)$. The windowing function has locality property which means it is just active within a local region controlled by $\sigma_i$ (*i.e.* Gaussian kernel window). Then $l_i$ points are cropped around the active region and treated as a frequency-selective filter (usually $l_i > \sigma_i$). Existing filter bank differs in their way of parameter setting, including filters' frequency response distribution, filter length length and windowing function. In classic hand-engineered features such as short-time Fourier transform (STFT) and LogMel feature, all these parameters are empirically chosen. Contrary to STFT, wavelet transform(Sturm, 2007) inversely correlates window length with frequency response so that the active region of the filter is internally decided by its frequency response.

**Time-Frequency Dynamic Resolution** Frequency resolution indicates the ability of discerning two adjacent frequency bins, time resolution corresponds to the capability of precisely localizing a sound source in time domain. The resolution in time and frequency domain jointly influence spatial localization precision because we encode spatial location on top of on pre-extracted time-frequency representation. According to Uncertainty Principle, however, the frequency resolution $\Delta_f$ and time resolution $\Delta_t$ satisfies $\Delta_f \cdot \Delta_t \geqslant C$ ($C$ is a constant). It means we cannot get optimal resolution in both time and frequency domain at the same time, but rather keep a trade-off between them: increasing the resolution in one domain inevitably sacrifices the resolution in the other domain.

Traditional hand-engineered sound feature's time-frequency resolution map is fixed because all filter bank construction relevant parameters are empirically chosen. Therefore, its time-frequency resolution map is evenly divided across both the time and frequency domain (see Fig.1 second last figure). Relaxing the frequency response and windowing length as trainable(Zeghidour et al., 2021) helps to achieve a better time-frequency resolution trade-off. The improvement is, however, limited because it uses one filter bank of the same filter length to process all sound signals (or perceive at one-scale). We categorize these traditional filter bank as syndistance filter bank to emphasize their equal temporal length property across all frequency responses.

In this paper, we rethink filter bank design and propose a synperiodic filter bank construction strategy. Our motivation is based on the fact that high-frequency sound signal can be adequately detected

with a short filter because its temporal length of one period is small. Conversely, lower frequency sound signals require wider filters. It thus shows using the same window size for all filters across the frequency range is not a good choice, a more desirable way is to use narrower window for filters with higher frequency responses. Under this guidance, we propose a synperiodic filter bank construction strategy. To better understand the main difference between syndistance filter bank and our proposed synperiodic filter bank, we can visualize them in complex-valued plane (see Fig.1, left-most), in which a filter is a complex exponential rotating in the complex plane counter-clockwisely, the rotating speed corresponds to the frequency it carries. In the complex-valued plane, all syndistance filters rotate to the same distance. Synperiodic filters, however, rotate to a predefined periodicity $\rho$, naturally resulting in narrow window for high-frequency filters and wide window for low-frequency filters.

Synperiodic filter bank lends us three advantages: it first avoids us setting window length for each filter separately, which is quite empirical and random; second, the constructed filter bank by design maintains a good time-frequency resolution trade-off; third, by simply varying the periodicity term $\rho$, we can easily obtain a group of synperiodic filter bank to process the raw waveform in multi-scale manner (as we will present in below). Our synperiodic filter bank construction strategy shares similar idea with Wavelet transform(Sturm (2007)) where it adopts a time shift and "squeezing ratio" to create achieve multi-scale perception, the difference is that we omit the time shift and instantiate the squeezing ratio with our proposed synperiodicity strategy. Moreover, synperiodic filter bank is multi-scale in time and frequency domain and seamlessly works well with convolution operation, and it is end-to-end trainable. Specifically, our proposed synperiodic filter bank can be represented as,

$$\mathcal{F}_{synp}^{\rho} = \{f_i(w_i, \rho, l_i) = \phi(w_i) \cdot \omega(w_i, \rho)\}_{i=1}^{M}, \quad where \quad \omega(\sigma_i) = \omega(w_i, \rho) \tag{2}$$

where we can see that the windowing function just depends on the center frequency $w_i$ and the periodicity term $\rho$. By varying the periodicity term $\rho$, we can obtain a group of synperiodic filter bank $\mathcal{F}_{synp} = \{\mathcal{F}^{\rho_1}, \mathcal{F}^{\rho_2}, \cdots, \mathcal{F}^{\rho_n}\}$, each group differs in its window size. The comparison between syndistance and synperiodic filter bank is shown in Fig. 1.

There are many ways to instantiate $\omega(w_i, \rho)$, as long as we guarantee the window length gradually reduces as the the frequency response increases. The simplest choice is to treat $\omega(w_i, \rho)$ as a constant, but we find it either results in too wide window for low-frequency filters or too narrow window for high-frequency filters. To mitigate this dilemma, we use logarithmic windowing function,

$$\omega(w_i, \rho) = 27 \cdot \log_{10}(w_i) - \rho, \quad \rho = \{-6, -11, -16\} \tag{3}$$

We set $\rho$ as $[-6, -11, -16]$ respectively to construct three synperiodic filter banks. The design of this window function is motivated by mel-scale frequency initialization strategy. By roughly setting a filter's bank width to be equal to the distance between its preceding and next frequency location in frequency domain, converting to time domain we can roughly get a logarithmic scale frequency-periodicity relationship (see Fig.4 in Appendix). In our implementation, synperiodic filter is created by multiplying a sinusoidal basis with learnable frequency response initialized in mel-scale by a Gaussian window with learnable width initialized through the windowing function by Eqn.3. Please note that each synperiodic filter bank group is initialized with independent learnable frequencies and window length, they are independently updated during training stage.

### 3.3 Multi-Scale Learning in Time and Frequency Domain

We use the previously constructed synperiodic filter bank group to convolve with mono-channel sound waveform with the same step size and padding strategy, resulting in the same size output for each single synperiodic filter bank. Since different filter group has different window size, we naturally achieve multi-scale learning in time domain. It maximally avoids us empirically selecting one window scale $\rho$ which might not be optimal, but instead uses a group of filter bank to enforce the neural network to strike a better time-frequency resolution trade-off in a data-driven way.

We further propose a strategy to enable multi-scale learning in frequency domain. The strategy is hierarchical: given a raw sound waveform with sampling frequency $F_S$, synperiodic filter bank's frequency is initialized within the range $[0, \frac{F_S}{2}]$ under Nyquist sampling theorem. If we downsample the sound waveform by a factor of 2, the resulting waveform can be processed by the lower-half

Figure 2: Multi-scale learning in frequency domain. Given the raw one channel sound waveform and pre-constructed synperiodic filter bank, we consecutively downsample the waveform by factor 2x, the newly downsampled waveform is processed by low-half filter bank from the proceeding filter bank. We can obtain time-frequency representation for on each frequency scale. These time-frequency representations share the same time length by adjusting step size. The final time-frequency representation is obtained by max-pooling them together.

filters in each group whose frequency response lies in $[0, \frac{F_S}{4}]$. This process that 2x-downsampling the waveform further process the downsampled waveform with filters with lower-half frequency response can potentially iterate a couple of times (in our case three times), resulting in multi-scale perception in frequency domain. The figurative illustration is shown by Fig. 2. Multi-scale learning in frequency domain brings us two extra benefits: 1) from data augmentation perspective, hierarchical 2x downsampling inevitably creates more waveforms, which matters a lot when labelling audio data is technically hard and time-consuming. 2) from the causality perspective, the adjacent 2x-downsampling strategy leads to dilated convolution for lower-frequency filters, because applying a filter to convolve with a downsampled waveform equals to convolve on the original waveform with dilated convolution (skip-2 connection). The resulting wider field of view (FoV) for lower frequency filters enables to learn better sound causality along the time axis (see Appendix A.4). In sum, by using learnable synperiodic filter bank group to process the raw waveform in multi-scale manner, we achieve a dynamic time-frequency resolution map that naturally maintains a better time-frequency resolution map fitting for sound source detection in a data-driven way.

**Computational Analysis** Synperiodic filter bank group introduces very few parameters (less than 1%) because they are parameterized filters. The trainable parameter number increases linearly w.r.t. synperiodic filter bank number. Their convolution with raw waveforms can also be efficiently implemented with 1D convolution.

### 3.4 Tansformer-like Backbone with Two Soft-Stitching Branches

Synperiodic filter bank group generates feature representation of shape $[T_0, N_0, C]$ ($T_0$ is much smaller than the $T$ but is still larger than $T_1$. $N_0$ equals to the filter number and we use 256, $C$ is channel number). Jointly learning framewise sound source semantic label and spatial location representation is a multi-task problem(Kendall et al., 2018; Misra et al., 2016). We propose a Transformer-like backbone with two paralleling and identical branches to learn each sub-task separately. To enforce information communication, we add a layerwise information exchange module: for the intermediate semantic label feature $f_s^i$ and spatial location feature $f_g^i$ learned by the $i$-th block, a learnable weight $W_i$ is introduced to linearly combine them together to get an updated $f_s^i$ and $f_g^i$ before feeding them to the next layer, $[f_s^i, f_g^i] = W_i \cdot [f_s^i, f_g^i]$. In practice, the Transformer-like backbone gradually reduces the learned feature's temporal length but increases the channel dimension accordingly. Finally we can get the final sound source representation of shape $[T_1, N]$. On top of the representation, we add trackwise permutation-invariant training (PIT) strategy(Cao et al., 2021) to trains the whole neural network in an end-to-end manner. The permutation invariant training strategy helps to address the polyphonicity challenge because it encapsulates a potential sound source's semantic label and spatial location within a track, so it won't be influenced by other sound sources.

## 4 Experiments

We conduct experiments on two tasks: direction of arrival (DoA) and physical location estimation. DoA estimation requires to decide arrival angle of sound source to microphone array. Physical location

Table 1: Evaluation result for DoA task. For Segment-based evaluation, we report detection error $ER_{20°}$, F measure $F_{20°}$ under DoA threshold $20°$, and classification dependent localization error $LE_{CD}$ and localization recall $LR_{CD}$. For event-based evaluation, we report mAP/mAR. The "Input" column labels are: 0. Raw waveform, 1. Log-Mel, 2. GCC-Phat, 3. Intensity Vector.

| Methods | Params | Input | Segment-Based Evaluation | | | | Event-based Evaluation | |
|---|---|---|---|---|---|---|---|---|
| | | | $ER_{20°}(\downarrow)$ | $F_{20°}(\uparrow)$ | $LE(\downarrow)(°)$ | $LR(\uparrow)$ | mAP($\uparrow$) | mAR($\uparrow$) |
| SELDNet(foa)(Adavanne et al., 2018) | 0.5 M | 1,3 | 0.63 | 0.46 | 23.1 | 0.69 | 0.087 | 0.152 |
| SELDNet(mic)(Adavanne et al., 2018) | 0.5 M | 1,2 | 0.66 | 0.43 | 24.2 | 0.66 | 0.079 | 0.140 |
| EIN(foa)(Cao et al., 2021) | 26.0 M | 1,2 | 0.30 | 0.77 | 8.9 | 0.84 | 0.134 | 0.256 |
| SoundDet(foa)(He et al., 2021) | 13.0 M | 0 | 0.25 | 0.81 | 8.3 | 0.82 | 0.197 | 0.294 |
| UTSC-Iflytek(foa+mic)(Wang et al., 2020) | Ensemble | 1,2,3 | 0.20 | 0.85 | 6.0 | 0.89 | - | - |
| SoundSynp(mic) | 60.0 M | 0 | 0.19 | 0.86 | 5.5 | 0.91 | 0.210 | 0.313 |
| SoundSynp(foa) | 60.0 M | 0 | **0.16** | **0.88** | **4.3** | **0.93** | **0.232** | **0.327** |

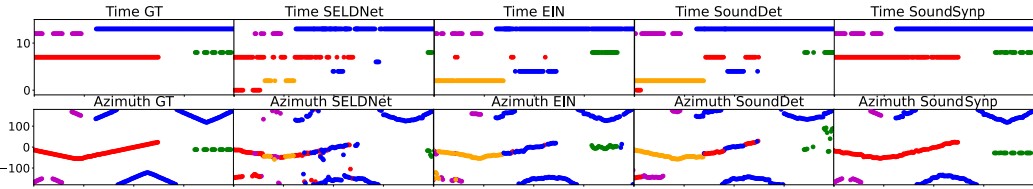

Figure 3: Qualitative comparison on DoA task. We show detected sound source temporal location (top row) and azimuth (bottom row). The horizontal axis is time, the vertical axis is semantic label (top) and azimuth in degree(bottom). Different color indicates different sound source category.

directly localizes a sound source's physical location (*i.e.* $(x, y, z)$ coordinates). For DoA estimation task, we use DCASE2020 sound event detection and localization dataset(Politis et al., 2020). It contains 14 sound sources with azimuth range $[-180°, 180°]$ and elevation range $[-45°, 45°]$. Two recording formats are available: FOA and MIC-array (refer to Appendix for detailed discussion). For more details about this dataset, please refer to (Politis et al., 2020). For physical location estimation, we use pyroomacoustics(Scheibler et al., 2018) simulator to simulate the dataset.

### 4.1 DIRECTION OF ARRIVAL ESTIMATION

We adopt two evaluation metrics: segment-based and event-based metric. Segment-based metric is a widely adopted evaluation metric(Adavanne et al., 2018; Cao et al., 2021), it couples semantic label and spatial location together: a semantic-correctly detected sound source needs to be spatially close enough to its ground truth location in order to be regarded as a true positive detection. Event-based based metric is newly proposed by(He et al., 2021) to comprehensively evaluate under different confidence scores. Like object detection from images(Lin et al., 2014), it computes mean average precision (mAP) and mean average recall (mAR) score.

We call our framework SoundSynp and compare it with four most recent methods: SELD-Net(Adavanne et al., 2018), EIN(Cao et al., 2021), SoundDet(He et al., 2021) and Utsc-Iflytek(Wang et al., 2020). SELDNet is the baseline model and it jointly trains sound source's semantic label and spatial location with a convolutional recurrent neural network (CRNN)(Chung et al., 2014). EIN(Cao et al., 2021) is a very recent work. It adopts multi-heads self-attention(Vaswani et al., 2017) to model temporal dependency and trackwise permutation-invariant training to train the model. SoundDet(He et al., 2021) directly learns from raw waveform with MaxCorr kernels, followed by an encoder-decoder neural network. Utsc-Iflytek(Wang et al., 2020) is ranked first in DECASE2020 challenge leaderboard[1], it combines MIC and FOA features and ensembles different models like ResNet(He et al., 2016a) and Xception(Chollet, 2017) to give final prediction.

**Implementation Detail** The network architecture is shown in table 4 in Appendix material. To train the neural network, we evenly divide the one minute long four-channel raw waveform into non-overlapping 4s short snippets. The raw waveform is first normalized to $[-1, 1]$ before feeding the neural network. We adopt Adam optimizer(Kingma & Ba, 2015) with an initial learning rate 0.0002 in the first 100 epochs and 0.00007 in the following 50 epochs. Batchsize is 16. The loss combination weight between classification head and regression head is $1 : 2$. During training, data

---

[1]see this link for leaderboard report.

augmentation method SpecAugment(Park et al., 2019) is applied. For DoA task, we regress direction of arrival angle in Cartesian coordinates $[x, y, z]$. In synperiodic filter bank groups, the filter length is 1025, each group's filter number is 256 and the step size is 600. Particularly, we have observed the initialized learnable synperiodic filter banks update its parameters intensively during the very early several epochs, and then gradually becomes stable. The final learned parameters are close to their initialization, one group's learned parameters (like frequency response and filter window length) is different from those learned by other groups, although they are initialized as the same in the start. We train each model five times independently and report the average score. The standard deviation is within 0.04 (for recall) and $0.2°$ for angle, 0.003 for mAP and mAR.

The result is given in Table 1, from which we see that SoundSynp achieves the best performance over all comparing methods by a large margin, under both segment-based and event-based metrics. Both SELDNet, EIN and UTSC-Iflytek use pre-extracted hand-engineered sound features, such as Logmel, GCC-Phat and Intensity Vector. SoundDet(He et al., 2021) and SoundSynp are the only two methods that directly learn from raw waveforms. At the same time, SoundSynp obtains better performance on FOA than MIC format, the same phenomena has been observed by all other methods. It thus shows FOA better fits for sound source detection than MIC. It is worth noting that Utsc-Iflytek(Wang et al., 2020) ensembles different powerful image-based 2D models to detect sound sources. However, our proposed SoundSynp still outperforms Utsc-Iflytek by a large margin. We don't report of mAP/mAR value for Utsc-Iflytek because it is a complex system and no detail about their system is available.

**Ablation Study** To disentangle the individual contribution of each part of our SoundSynp framework to the whole performance improvement, we conduct three ablation studies. **First**, the individual contribution of Synperiodic filter bank. We replace hand-engineered sound feature head of SELDNet, EIN and learnable MaxCorr filter bank with our proposed Synperiodic filter bank to test their corresponding performance. It helps to remove the influence of the backbone neural network of different models and thus helps to get direct comparison of synperiodic filter bank with other front-end. **Second**, we replace SoundSynp's synperiodic filter bank group with widely-used MFCC(Davis & Mermelstein, 1980) and Log-Mel that is used by SELDNet(Adavanne et al., 2018) and EIN(Cao et al., 2021), respectively. It disentangles synperiodic filter bank group with Transformer-like backbone, thus helps to figure out if the performance gain is simply brought by backbone network. **Third**, internally, we test five synperiodic variants: (1) synperiodic filter bank with just multi-scale perception in frequency domain (SoundSynp_MSFreq), (2) just multi-scale perception in time domain (SoundSynp_MSTime), (3) Synperiodic filter bank with frequency responses linearly initialized in Nyquist frequency range (SoundSynp_Linear, compare with our mel-scale initialization), (4) just one synperiodic filter bank without multi-scale perception neither in time nor frequency domain (SoundSynp_SingleScale). (5) synperiodic filter bank with rectangular band-pass frequency response initialization (SoundSynp_Sinc), like SincNet(Ravanelli & Bengio, 2018) does. The internal comparison helps us to figure out the necessity of each part of synperiodic filter bank design.

The ablation study result is shown in Table 2. We can observe that: **First**, using synperiodic filter bank as a replacement of existing filter bank can help to improve the corresponding performance. **Second**, replacing SoundSynp's synperiodic filter bank with classic hand-engineered features inevitably reduces the performance under all evaluation metrics, its thus shows learning from pre-extracted fixed and single-scale sound representation leads to inferior performance than our proposed multi-scale synperiodic filter bank. **Third**, the absence of multi-scale perception in either frequency domain or time domain inevitably reduces the performance. We find sound source semantic label detection suffers more in single-scale perception in time domain than in frequency domain (see better performance on $ER_{20°}$, and $F_{20°}$ score), which shows frequency domain multi-scale perception is vital for semantic label estimation. Similarly, we can observe that multi-scale perception in time domain is vital for sound source spatial location estimation (see better performance on $LE$ and $LR$ score). Linearly initialized filter bank frequency response reduces the performance, which shows assigning more filters to the lower frequency range is important. But this conclusion might be data-dependent because we find DCASE dataset contains many low-frequency sounds like burning fire and footsteps. Moreover, reducing the synperiodic filter bank group to one groups with just single-scale perception leads to comparably the worst performance, it thus shows multi-scale perception in both time and frequency domain is essential for DoA-based sound source detection. Lastly, SoundSynp_Sinc leads to slightly inferior performance than our used mel-scale initialization strategy, it shows our proposed synperiodic filter bank is a general filter bank that can be adopted to other frequency sensitive filter bank.

Table 2: Ablation study report. Internally, we replace SELDNet, EIN and SoundDet head with synperiodic filter bank (denoted as _Synp). Externally, we test different modified SoundSynp versions.

| Methods | Blocks | Segment-Based Evaluation | | | | Event-based Evaluation | |
|---|---|---|---|---|---|---|---|
| | | $ER_{20^\circ}(\downarrow)$ | $F_{20^\circ}(\uparrow)$ | $LE(\downarrow)(^\circ)$ | $LR(\uparrow)$ | mAP($\uparrow$) | mAR($\uparrow$) |
| SELDNet(foa)(Adavanne et al., 2018) | Conv2D, biGRU | 0.63 | 0.46 | 23.1 | 0.69 | 0.087 | 0.152 |
| SELDNet_Synp(foa)(Adavanne et al., 2018) | | 0.57 | 0.49 | 22.1 | 0.75 | 0.093 | 0.165 |
| SELDNet(mic)(Adavanne et al., 2018) | Conv2D, biGRU | 0.66 | 0.43 | 24.2 | 0.66 | 0.079 | 0.140 |
| SELDNet_Synp(mic)(Adavanne et al., 2018) | | 0.61 | 0.48 | 23.0 | 0.70 | 0.090 | 0.153 |
| EIN(foa)(Cao et al., 2021) | Conv2D, MHSA | 0.30 | 0.77 | 8.9 | 0.84 | 0.134 | 0.256 |
| EIN_Synp(foa)(Cao et al., 2021) | | 0.27 | 0.79 | 8.3 | 0.87 | 0.138 | 0.258 |
| SoundDet(foa)(He et al., 2021) | Conv1D, LSTM | 0.25 | 0.81 | 8.3 | 0.82 | 0.197 | 0.294 |
| SoundDet_Synp(foa) (He et al., 2021) | | 0.22 | 0.83 | 7.9 | 0.85 | 0.204 | 0.299 |
| SoundSynp_MFCC(foa) | Conv2D, MHSA | 0.21 | 0.83 | 6.9 | 0.85 | 0.203 | 0.301 |
| SoundSynp_LogMel(foa) | | 0.22 | 0.84 | 6.8 | 0.85 | 0.207 | 0.308 |
| SoundSynp_MSFreq(foa) | Conv2D, MHSA | 0.20 | 0.85 | 7.3 | 0.86 | 0.210 | 0.284 |
| SoundSynp_MSTime(foa) | | 0.22 | 0.82 | 7.0 | 0.88 | 0.172 | 0.282 |
| SoundSynp_Linear (foa) | | 0.21 | 0.83 | 8.8 | 0.83 | 0.168 | 0.259 |
| SoundSynp_SingleScale(foa) | | 0.25 | 0.80 | 9.0 | 0.81 | 0.164 | 0.246 |
| SoundSynp_Sinc(foa) | | 0.19 | 0.86 | 5.2 | 0.91 | 0.226 | 0.316 |
| SoundSynp(foa) | Conv2D, MHSA | **0.16** | **0.88** | **4.3** | **0.93** | **0.232** | **0.327** |

Table 3: Physical location estimation result report. We set distance threshold $1.0m$, which means a detected sound sources has to lie within this threshold in order to be treated true positive.

| Methods | Segment-Based Evaluation | | | | Event-based Evaluation | |
|---|---|---|---|---|---|---|
| | $ER_{1.0m}(\downarrow)$ | $F_{1.0m}(\uparrow)$ | $LE(\downarrow)(m)$ | $LR(\uparrow)$ | mAP($\uparrow$) | mAR($\uparrow$) |
| EIN(Cao et al., 2021) | 0.45 | 0.71 | 0.71 | 0.73 | 0.114 | 0.240 |
| EIN_Synp(Cao et al., 2021) | 0.45 | 0.72 | 0.67 | 0.75 | 0.117 | 0.244 |
| SoundDet(He et al., 2021) | 0.43 | 0.67 | 0.68 | 0.74 | 0.118 | 0.271 |
| SoundDet_Synp(He et al., 2021) | 0.44 | 0.69 | 0.64 | 0.79 | 0.123 | 0.271 |
| SoundSynp | **0.35** | **0.79** | **0.53** | **0.83** | **0.154** | **0.279** |

One qualitative comparison is shown in Fig. 3. We can clearly see that SELDNet generates mixed prediction at different time steps and DoA locations. SoundDet and EIN give non-existing sound sources (orange color). When multiple sound sources happen at the same time (polyphonicity), SoundDet and EIN are easily failed to predict the right spatial location (discretized blue and red color). Our method predicts more spatially and temporally consistent sound sources by maximally keeping sound source's continuity and consistency.

## 4.2 PHYSICAL LOCATION ESTIMATION

We use Pyroomacoustics simulator(Scheibler et al., 2018) to simulate a shoebox like room with $[7m, 5m, 3m]$ size. Four microphones (24kHz sampling rate) are configured in a plane with $5cm$ mutual distance and put in the room center ($[3.5m, 2.5m, 1.5m]$). We choose three commonly heard sounds from BBC sound effect site[2]: cat meowing, baby talking and dog barking. All these seed sounds last less than 10s. During simulation, we put all seed sound in the plane $z = 1.5m$ (we constrain to a plane because we find relaxing $z$ easily leads EIN model to get stuck in a local minima like $[-1, -1, -1]$), but with $x$ randomly in $[2m, 5m]$ and $y$ randomly in $[1m, 4m]$, up to two sounds can happen at the same time. In sum, we have simulated 500 one-minute recordings (400 for train, 100 for test), with an average of four sounds recorded per recording. For physical location prediction, we regress the angle and range. The result is in Table 3. We do not report SELDNet result as we find it doesn't converge during train. We can see from the table that SoundSynp produces the best performance than over EIN and SoundDet network. Replacing their front-end with synperiodic filter bank also improves the performance. It thus shows our proposed SoundSynp is capable of learning representation for physical-location based sound source detection.

## 4.3 DISCUSSION AND CONCLUSION

We have proposed a novel learnable sound feature extractor front-end: synperiodic filter bank. It requires very few parameters and is computational efficient. It extracts feature in multi-scale manner in both time and frequency domain. Although it shows excellence in detecting sound sources, we believe it can be used as a general front-end to tackle other sound tasks, like speech processing and keyword spotting.

---

[2]see https://sound-effects.bbcrewind.co.uk/

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

# A  APPENDIX

## A.1  SPATIAL LOCATION ENCODING IN FREQUENCY DOMAIN AND RECORDING FORMAT DISCUSSION

We discuss the detailed spatial location encoding for FOA and MIC sound waveform recording format. The spatial location encoding is based on 2D feature learned by the Gabor filter bank for each sound waveform channel, which can be represented as $\{F_i = (R_i, I_i)\}_{i=1}^4$. $R_i$ and $I_i$ are real part and imaginary part feature of the $i$-th channel sound waveform, respectively.

FOA format is well-known as first-order Ambisonics (B-format). It contains four channels: omni-directional, $x$-directional, $y$-directional and $z$-directional components, respectively. The instantaneous sound intensity vector is often used as spatial location (or phase difference) feature, which can be computed through the cross-spectrum between the omni-directional channel to the remaining $x, y, z$-directional. As a result, we have obtained 3 channel spatial location encoding feature.

$$IV_x = F_0^* \cdot F_1, \;\; IV_y = F_0^* \cdot F_2, \;\; IV_z = F_0^* \cdot F_3 \tag{4}$$

where $F_0^*$ indicates the conjugate of the omni-directional feature. The three cross-spectrum feature $IV_x$, $IV_y$ and $IV_z$ are stacked together and further normalized before serving as the spatial encoding feature.

MIC format is well-known as tetrahedral microphone array. The four microphones are mounted in spherical coordinates with four distinct orientations. We treat the four microphones equally and compute the phase difference between any two microphones. Thus a total of six channels spatial location feature can be constructed. Specifically, we choose to compute GCC-PHAT(Brandstein & Silverman, 1997) like cross-spectrum feature. For any two channel $m$ and $n$, we compute the angle between the real part and imaginary part of the cross-spectrum.

$$SL = angle(F_m^* \cdot F_n), \; m \neq n, m = 1, 2, 3, 4; n = 1, 2, 3, 4 \tag{5}$$

$SL$ indicates the spatial location feature computed by the sound waveform channel $m$ and $n$. The $angle(\cdot)$ equals to a frequency amplitude normalization operation, like the GCC-PHAT(Brandstein & Silverman, 1997) does. Please note that all the spatial location feature computation operations are differentiable so the whole neural network becomes end-to-end trainable.

## A.2  NETWORK ARCHITECTURE AND TRAINING DETAILS

SoundSynp neural network architecture is given in Table 4. To train the neural network, we evenly divide the one minute long four-channel raw waveform into non-overlapping 4s short snippets. The raw waveform is first normalized to $[-1, 1]$ before feeding the neural network. We adopt Adam optimizer(Kingma & Ba, 2015) with an initial learning rate 0.0002 in the first 100 epochs and 0.00007 in the following 50 epochs. Batchsize is 16. The loss combination weight between classification head and regression head is $1 : 2$. During training, data augmentation method SpecAugment(Park et al., 2019) is applied. For DoA task, we regress direction of arrival angle in Cartesian coordinates $[x, y, z]$. In synperiodic filter bank groups, the filter length is 1025, each group's filter number is 256 and the step size is 600. Particularly, we have observed the initialized learnable synperiodic filter banks update its parameters intensively during the very early several epochs, and then gradually becomes stable. The final learned parameters are close to their initialization, one group's learned parameters (like frequency response and filter window length) is different from those learned by other groups, although they are initialized as the same in the start.

## A.3  SYNPERIODIC FILTER BANK FREQUENCY-PERIODICITY RELATIONSHIP DETERMINATION

Mel-scale time-frequency representation has been widely used in both traditional sound feature like MFCC(Davis & Mermelstein, 1980), LogMel and learnable filter bank(Zeghidour et al., 2021). It initializes the filter bank in frequency domain, in which high-frequency filter has wider window length. We transform the filter bank into time domain and can naturally get a roughly logarithmic-scale frequency-periodicity relationship, in which narrower window width is associated with high-frequency

Table 4: SoundSynp neural network architecture. The layer follow $name@kernelsize, stride$ format, and synperiodic filter bank follow $name@kernelsize, stride, groups$ format. FC is fully connection layer, AvgPool is the average pooling layer, MaxPool is max-pooling layer. B is the batch-size, T is input waveform time-length. All convolution layers are followed by a batch normalization layer and Relu activation layer. We represent the neural network architecture on FOA recording format with sampling rate 24k Hz and label resolution 100 ms. It can be easily adjusted to fit other cases. Please note that since the backbone neural network has two identical branches, we just show one branch here.

| layer | filter num | output size |
|---|---|---|
| Input: [B,4,T] | | |
| Synperiodic Filter Bank Groups | | |
| SynperiodicFilterBank@1024,600,3 | 256 | [B, 256, T/600, 21] |
| Backbone Conv block1 | | |
| Conv2d@3,1 | 128 | [B, 256, T/600, 128] |
| Conv2d@3,1 | 128 | [B, 256, T/600, 128] |
| AvgPool@2,1 | None | [B, 128, T/600, 128] |
| Backbone Conv block2 | | |
| Conv2d@3,1 | 256 | [B, 128, T/600, 256] |
| Conv2d@3,1 | 256 | [B, 128, T/600, 256] |
| AvgPool@2,1 | None | [B, 64, T/600, 256] |
| Backbone Conv block3 | | |
| Conv2d@3,1 | 256 | [B, 64, T/600, 256] |
| Conv2d@3,1 | 256 | [B, 64, T/600, 256] |
| AvgPool@2,1 | None | [B, 32, T/600, 256] |
| Backbone Conv block4 | | |
| Conv2d@3,1 | 512 | [B, 32, T/600, 512] |
| Conv2d@3,1 | 512 | [B, 32, T/600, 512] |
| AvgPool@2,1 | None | [B, 16, T/600, 512] |
| Backbone Conv block5 | | |
| Conv2d@3,1 | 512 | [B, 16, T/600, 512] |
| Conv2d@3,1 | 512 | [B, 16, T/600, 512] |
| AvgPool@16,1 | None | [B, T/600, 512] |
| Backbone MHSA block1 | | |
| MHSA@8,1024 | 512 | [B, T/600, 512] |
| AvgPool@2,1 | None | [B, T/1200, 512] |
| Backbone MHSA block2 | | |
| MHSA@8,1024 | 512 | [B, T/1200, 512] |
| AvgPool@2,1 | None | [B, T/2400, 512] |
| Backbone MHSA block3 | | |
| MHSA@8,1024 | 512 | [B, T/2400, 512] |
| FC | class num | [B, T/2400, class num] |
| FC | class num x 3 | [B, T/2400, class num x 3] |
| Trackwise Permutation Invariant Head | | |
| Multi-label Classification | None | scalar |
| Location Regression | None | scalar |

filters. We thus set $\omega(w_i, \rho) = 27 \cdot \log_{10}(w_i) - \rho$. We plot our synperiodic filter bank window function and the mel-scale initialized windowing function in Fig. 4, it shows our proposed windowing function naturally approximates the mel-scale windowing function.

## A.4 CAUSALITY IN MULTI-SCALE FREQUENCY DOMAIN PERCEPTION

Fig.5 shows how causality is achieved in multi-scale frequency domain perception.

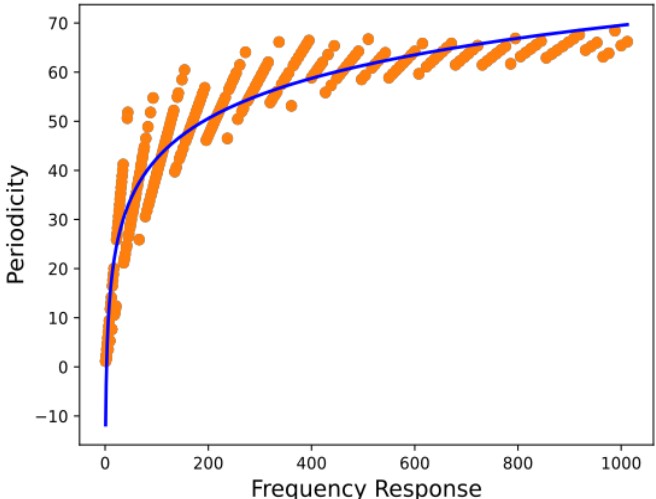

Figure 4: The relationship between filter frequency response and the periodicity. Green curve: our proposed windowing function. Light orange dots: mel-scale initialized frequency-periodicity relationship.

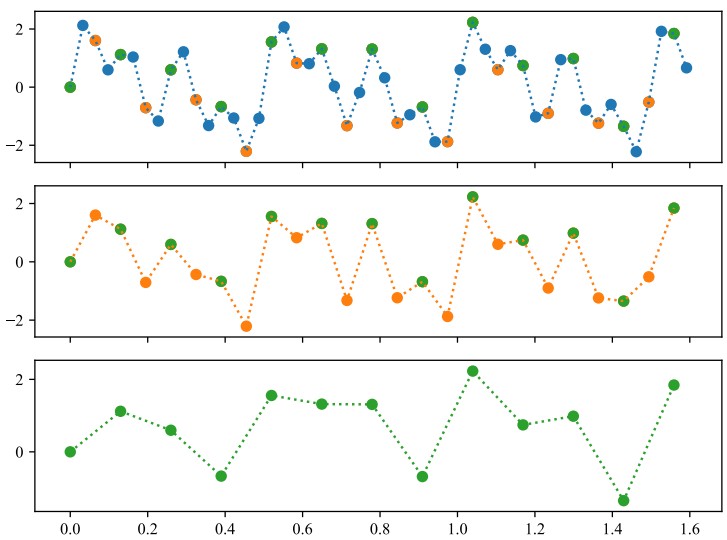

Figure 5: Consecutively downsampled waveform has longer time range in the original waveform (top row). So processing downsampled waveform increases the Causality.

## A.5 COMPUTATION TIME REPORT

The computation time of all comparing methods is given in Table 5, from which we can see that our proposed SoundSynp framework requires between 2x-3x more computation time than SELDNet, EIN and SoundDet due to its large Transformer-like backbone. However, SoundSynp computation time is still within a controllable range.

Table 5: Inference time on Intel(R) Core(TM) i9-7920X CPU. The waveform pre-processing time is contained for SELDNet and EIN. Each individual computation time is computed by averaging 100 independent inference of 4s sound waveform.

| SELDNet | EIN | SoundDet | SoundSynp |
|---------|--------|----------|-----------|
| 1.20 s | 2.20 s | 1.25 s | 3.1 s |

