# OpenReview forum: "Sound Source Detection from Raw Waveforms with Multi-Scale Synperiodic Filterbanks"
_ICLR.cc/2022/Conference — ICLR 2022 Submitted_

### Official Review · Reviewer_dYuW · 2021-10-31

**Correctness:** 3
**Technical Novelty And Significance:** 3
**Empirical Novelty And Significance:** 2
**Recommendation:** 6
**Confidence:** 5

**Main Review:**

Despite interesting presented results with the proposed approach, this paper contains several major flaws.
For all the reasons reported below, I cannot recommend to accept this paper in the current form, however I encourage the authors to submit again an improved version of their work.

I) the overall organization of the paper should be improved. First,  it contains a lot of references to previous work for which the link with the present paper is not very clear (cf. Section 2). I recommend to formulate the problem before discussing the existing approaches to explain in what the new proposed method is new and significant. I also suggest to add a formal and clear definition of "synperiodicity" that is missing since it is not a standard concept in signal processing theory. This point is important  since it is a possible new contribution claimed by the authors. For the moment, I understand that "synperiodic" filter-bank is a reformulation of well-known concepts already rigorously presented in time-frequency analysis books such as "P. FLANDRIN, Time-frequency/time-scale analysis. Academic press, 1998". or "S. MALLAT, A wavelet tour of signal processing. Elsevier, 1999."  The authors should explain rigorously what is "synperiodicity" and "syndistance" with consideration with previously established theoretical results. The handled concepts such as time-frequency resolution are currently too superficially presented in the actual form of the paper.

II)The chosen mathematical notations are not always adequate. For example from Eq. (1), the goal of sound source detection is to estimate the number of sources K with their parameters of the sources S. Here, the link between the representation $\mathcal P$ and $S$ is not clear. Moreover, source detection and tracking is a problem not directly related to deep neural network that is only a possible approach to address this problem. I suggest to the author the use a standard and general introduction to the problem as presented in previous works to make the link.  Looking in detail, the formal definition of what is $sp_k$, $sl_k$ is not very clear, this is also the case for several parameters implicitly introduced in the paper such as $\sigma_i$, $\rho$, etc. The other equations and the following notations are not very rigorous too (eg. Eqs. (2)- (5) ) and don't help to clarify the understanding of the method and to make the link with standards tools such as the Fourier transform. For example, one should guess to what refers the frequency response $\phi(\omega_i)$ multiplied by the window function  $w$ that should be a function of time and should be parametrized by a width $\sigma_i$... In Eq.(5) the window now becomes a function of $\omega_i$ and $\rho$ that is no more consistent.
To fix that, I suggest to the authors to  only describe in detail their own contribution mainly summarized in the neural network architecture presented in Table 4 and to discuss the changes in comparison to similar deep neural-based approaches with dedicated simulations or theoretical results to motivate their choices. For example, I would be interested in a accurate comparison of the resolution of the proposed method with several existing TF representation such as STFT or CWT using different analysis windows.

III) Despite good claimed results, the experiments are not sufficiently detailed to be reproduced. What is exactly the evaluation protocol here ? What are the set of training/validation and test ? The computation time ?  Do the authors have reimplemented the compared state-of-the-art methods or did they only reported results from the literature ? Can please provide more details about the implementation of your method, used language and framework, providing implementation considerations and/or with source code ? Can you please explain how Fig. 4 and 5 are computed ? Can you please detail what are the axis with the proper units in all the figures ? Moreover, the authors should explain the evaluation metrics with a fair discussion with the other methods. What is the motivation of the ablation study? This  problem of interpreting the results is related to my Section II) where the authors should explain why their method perform better, discuss several relevant points with a consideration of the time-frequency resolution of the proposed "synperiodic" filterbank claimed as the main new contribution.

IV) The paper could also benefit of a major rewriting and English spell-check to fix typos and grammatical errors.

**Summary Of The Paper:**

This paper proposes a method for improving the detection and the localization of several sound sources using a 4-channel signal (The addressed configuration, determined or under-determined case is not detailed in this paper).
The proposed approach is based on a so-called "synperiodic" filter bank representation that is used as the input of a deep convolutional neural network. Finally method proposed by the authors is comparatively evaluated on the DCASE2020 dataset and pretends to provide the best results.



**Summary Of The Review:**

The authors address a very interesting topic considering the computation of the suitable signal representation as the input of a deep neural network to address the problem of estimating the signal parameters of interest.
Despite the presented "good" results, the experiments are not reproducible since relevant information related to the protocol and the implementation are missing. Moreover, the theoretical concepts introduced in this study to describe the main contributions from the authors  are too superficially presented.
Hence, I recommend to reject this paper in present state but I strongly encourage the authors to resubmit a more clear and more transparent study (for example to DCASE 22) to introduce their possibly novel approach.

---

> ### Author Response · Authors · 2021-11-15
> **Reply**
>
> **Q1**: The overall organization of the paper should be improved.
>
> **A1**:
>
> 1. Related Work section gives a complete review of a) how existing approaches tackle sound source detection challenge. b) current research on design neural network consuming sound raw waveforms. c) multi-scale representation related research. It helps readers to have a better understanding of past work before reading our work.
>
> 2. In our revised version, we combined "sound source problem definition" and "sound source clues" section together. In Sec. 3.2 (Sec. 3.3 in the first version), we modified Eqn. (2) and Eqn. (3) to better illustrate synperiodic filter bank idea. We gave detailed introduction of synperiodic filter bank in this section. In a nutshell, syndistance filter bank indicates traditional sound feature like MFCC, STFT where the filter length is the same across all frequency bins. Syndistance is used to emphasize the *same filter length* property. On the contrary, one way to better understand **synperiodic** is to treat frequency as rotating speed in the complex plane (Fig. 1). Since high-frequency filter rotates faster than low-frequency filter, requiring them to the same periodicity (in logarithmic scale in Eqn (3)) resulting in narrow filters carrying high frequency and wide filters carrying low frequency. Synperiodic filter bank naturally maintains a good time-frequency resolution, it can be further optimized in a data-driven way during the training stage.
>
> 3. The book "A wavelet tour of signal processing" introduces every aspect of sound processing. As far as we know, syndistance filter idea mainly derives from Fourier transform realm, in which the resulting time-frequency resolution is fixed and cannot be changed (chapter II, IV in the textbook). In wavelet multi-scale representation, a filter is controlled by two independent factors: a squeezing factor and time shift factor. We can get a bunch of such filters by varying the two factors. Synperodic filter bank is, however, designed to be compatible with the way 1D conv. operates:
>
> * the time shift factor is omitted because we convolve with the raw waveform by scanning from the left to right with pre-defined stepsize.
>
> * Synperiodic filter correlates squeezing rate with the frequency each filter carries, we instantiate squeezing rate with periodicity.
>
> * In multi-scale perception in time domain, we vary the periodicity to obtain filters of different lengths but carrying the same frequency.
>
> * In multi-scale perception in frequency domain, we draw inspiration from Mallat wavelet design to use the same filter bank to process the 2x- downsampled waveform.
>
>  * In textbook, all parameters are fixed and not learnable. In our setting, all parameters are learnable and can be optimized during training. Each filter bank group params are adjusted independently.
>
> **Q2**: Math notations are not adequate.
>
> **A2**:
> 1. We formulate sound source detection as a framewise semantic label classification and spatial location regression task. So our goal is to learn to a framewise representation $\mathcal{P}$. SELDNet, EIN and UTSC-Iflytek use the same problem formuation. $sp_k$ and $sl_k$ are the $k$-th sound source's spatial location and semantic label respectively.
>
> 2. In our revised version, we rewrite Eqn. (5) and add another equation to explicitly show the relationship between $\sigma_i$, $w_i$ and $\rho$. We reorganized the paper in the revised version, as we said in Q1.
>
> 3. Moreover, to further figure out which part brings the improvement, we conducted three ablation studies. Please see "Rebuttal Summary" for more discussion on ablation study.
>
> **Q3**: Evaluation protocol and Experiments.
>
> **A3**:
>
> 1. Two evaluation protocols: segment-based (see [link](http://dcase.community/challenge2020/task-sound-event-localization-and-detection)) and event-based (see [link](http://proceedings.mlr.press/v139/he21b/he21b.pdf)), we gave introduction of them in the experiment sec..
>
> 2. We follow the official split-600 for train and 200 for test, refer to [link](https://arxiv.org/pdf/2010.13092.pdf).
>
> 3. The computation time is added in Appendix.
>
> 4. We use SELDNet official code [link](https://github.com/sharathadavanne/seld-dcase2020), EIN official code [link](https://github.com/yinkalario/EIN-SELD) . We use the UTSC-Iflytek reported score [link](http://dcase.community/challenge2020/task-sound-event-localization-and-detection) and implement SoundDet code.
>
> 5. In Fig. 4,  we use log-mel filter initialization ([link](https://github.com/google-research/leaf-audio)) to define the periodicity in logarithmic scale. Detailed discussion is in Eqn. (3) and the closing paragraph. We provide the code of Fig. 4 in our revised version. Fig. 5 shows the Causality brought by multi-scale perception in frequency domain: convolving on 2x- downsampled waveform increases temporal coverage in the original waveform.
>
> **Q4**: Paper writing.
>
> **A4**: We carefully revised the whole paper in our revised version.

---

### Official Review · Reviewer_DURm · 2021-11-01

**Correctness:** 3
**Technical Novelty And Significance:** 3
**Empirical Novelty And Significance:** 3
**Recommendation:** 6
**Confidence:** 2

**Main Review:**

**Note**: I am not an experienced researcher in the sound source detection. My reviews may miss some published results or technology.

## Strengths

+ The paper is clearly written, with sufficient technical details.
+ The novelty and motivation is valid and the proposed method achieves state-of-the-art performance.

## Weakness

- Does the synperiodic filterbank consider the physical characteristics of the sound receiver system ($e.g.$, spatially-deployed microphone array)? If the receiver system structure is changed, does the synperiodic filterbank still generalize well?

**Summary Of The Paper:**

This submission addresses the sound source detection problem. Compared to previous method, the authors propose a novel synperiodic filerbank compared to syndistance bank, which learns a data-dependent time-frequency resolution map. Experiments show that the proposed method achieves state-of-the-art performance on several datasets.

**Summary Of The Review:**

**Note**: I am not an experienced researcher in the sound source detection. My reviews may miss some published results or technology.

Overall, I think this submission has solid novelty and state-of-the-art performance on the sound source detection task. Thus, I am inclined to accept the submission. However, my comments may not be professional and I would wait for authors response before making the final decision.

---

> ### Author Response · Authors · 2021-11-15
> **Reply**
>
> Thanks for your comments.
>
> **Q1**: Does the synperiodic filterbank consider the physical characteristics of sound receiver system?
>
> **A1**: Our proposed synperiodic filterbank is designed to process mono-channel waveform. For inter-channel interaction, we encode the phase difference on the time-frequency representation extracted by synperiodic filterbank. The detailed discussion is presented in Appendix A.1. Since we use inter-channel phase difference as sound source spatial location clue, we require the four microphones are configured close to each other (e.g. 4 cm distant with each other) because there are other inter-channel information to be considered like amplitude difference and energy decay if microphone array is distantly-configured (e.g. put microphones at a large house's different corners).
>
> **Q2**: If the received system structure is changed, does the synperiodic filter bank still generalize well?
>
> **A2**: Synperiodic filter bank is proposed to extract mono-channel time-frequency representation in multi-scale manner in both time and frequency domain. Theoretically, it can be used as a general sound waveform time-frequency representation extractor to solve other acoustic tasks (although we did not test it in this paper). If the system structure is changed, we think synperiodic filter bank can still be used as sound feature extraction front-end. In this paper, we have tested synperiodic filter bank generalizes well on both direction-of-arrival (DoA) task and sound source physical location ($[x,y,z]$ coordinate) estimation task. It needs more test if the receiver configuration changes too much (i.e. rather than setting four microphones close with each other, but putting them far apart and at different corners)

---

### Official Review · Reviewer_TyRd · 2021-11-02

**Correctness:** 3
**Technical Novelty And Significance:** 2
**Empirical Novelty And Significance:** 1
**Recommendation:** 5
**Confidence:** 4

**Main Review:**

Strengths:
1. The premise of the paper that using a more appropriate Filterbank which learns to choose different time-frequency resolution tradeoffs based on data is indeed interesting. The authors have made a nice effort in figuring out the details and their experimental evaluation.
2. The ablation study of using the proposed "Synperiodic Filterbank" features with existing methods was definitely needed and insightful to show that features do carry more information than the traditional features.

Weakness:
1. Personally I think using a significantly larger backbone of transformers on the proposed features to show improvement was not a fair comparison. I would recommend the authors to also experiment with the same or similar Transformer architecture using traditional MFCC or LFBE features to clearly understand what brings more value to the table - the model architecture or the feature extractor. From the existing ablation study it feels like the feature extractor offers a modest improvement at best.
2. It appears that the authors have reported single / best model accuracy numbers and it is not clear how reliable these numbers are - perhaps reporting average metrics  + standard deviation across multiple model runs (with random initialization) would be better ?
3. Several related works concerning raw waveform based sound detection and separately raw waveform based source localization are missing and hence the related works section does not offer a balanced review of the problem space:
[1] T. Kim, J. Lee, and J. Nam, “Comparison and analysis of sample CNN architectures for audio classification,” IEEE Journal of Selected Topics in Signal Processing, vol. 13, no. 2, pp.285–297, May 2019. - Review of Raw-audio based architectures using CNN for event classification.
[2] J. M. Vera-Diaz, D.Pizarro, and J. M. Guarasa, “Towards end-to-end acoustic localization using deep learning: from audio signal to source position coordinates,” CoRR, vol. abs/1807.11094, 2018. Raw audio based single source localization
[3] H. Sundar, W. Wang, M. Sun and C. Wang, "Raw Waveform Based End-to-end Deep Convolutional Network for Spatial Localization of Multiple Acoustic Sources," IEEE International Conference on Acoustics, Speech and Signal Processing (ICASSP), 2020, pp. 4642 - 4646.
4. The paper can benefit greatly from re-writing the motivations and editing. A few examples of unnecessary claims not justified in the experiments and sentences which can benefit from revision are :
a) In the Abstract: "...Existing methods fail to do so because they either extract hand-engineered features (i.e. STFT, LogMel) that require a great deal of parameter tuning work (i.e. filter length, window size), or propose to learn a single filter bank to process sound waveforms in a single-scale that often leads to a limited time-frequency resolution capability" - I don't think the existing methods are "failing". They may not be performing as well as the model the authors of this paper have.
b) In multiple places the phrase - "filter’s length and frequency response are inversely related" is used. This is not making sense to me. Should this be "filter’s length and frequency resolution are inversely related" ?
c) Sentence needs revision:
    i) "Traditional sound features are human-auditory system biased."
   ii) "Sound source detection has been previously treated as sound structure (Thrun, 2006), sound object(He et al., 2021) and sound event detection and localization (SELD task)"
  iii) "Our motivation is based on the fact that high frequency sound signal can be adequately detected with a shorter filter, enlarging the filter length doesn’t necessarily increase the frequency resolution but inevitably reduces the filter’s time resolution." Amenable to wrong interpretation - Are the authors referring to bio-inspired processing where humans are less sensitive resolving between
  iv) "The design of this window function is motivated by the mel-scale frequency initialization strategy, in which two neighboring frequencies are more closely spaced in lower frequency part and higher frequency part" - not clear what this means.
  v) "The parameter number is just linear to filter number." - needs revision.
d) In Eq. (5) what is l (ell) ?


**Summary Of The Paper:**

This paper addresses the problem of sound event detection and localization from multi-channel raw audio waveforms. Essentially, a different audio front end feature extraction scheme - "Synperiodic Filterbank" if proposed which can be parameterized and jointly learnt along with a backbone classifier in an end-to end manner. The feature extraction scheme proposed is grounded in principle of uncertainty governing time-frequency resolution. Through ablation studies the authors show that this feature extraction scheme performs better (albeit marginally) compared to some of the traditionally used feature extraction schemes like MFCC, and LFBEs. Subsequently, the authors show that combining the proposed "Synperiodic Filterbank" with a significantly large model (~3X times the closest baseline model in terms of No. of trainable parameters) based on the Transformer Architecture offers a dramatic improvement in the results compared to the baseline.

**Summary Of The Review:**

Overall I think the authors address a curious idea. The presentation is not entirely widely accessible and can benefit greatly from revision. The experiments are interesting but not conclusive enough to understand the root cause of improvements.

---

> ### Author Response · Authors · 2021-11-15
> **Reply**
>
> **Q1**: Backbone network concern, fair comparison to figure out which part contributes most.
>
> **A1**: In the first version, we conducted two ablation studies:
>
> 1. First, replacing SELDNet, EIN and SoundDet feature extraction head with our proposed synperiodic filter bank, experimental results show that our proposed synperiodic filter bank can be used as a plug-and-play front-end sound feature extractor by exiting methods to improve their performance.
> 2. Second, we tested five variants of synperiodic filter bank, like without multi-scale perception in time or frequency domain. Experimental results show all the tested variants lead to inferior performance. It thus shows the necessity of each part of synperiodic filter bank design, especially the multi-scale in time domain ability by varying the periodicity and in frequency domain ability by convolving on 2x- downsampled waveforms.
>
> In our revised version, we conducted another ablation study: replacing our proposed SoundSynp's synperiodic filter bank head with traditional MFCC and LogMel (we guess it is the LFBE feature your have mentioned) sound feature. It turned out that removing synperiodic filter bank head reduces the performance, but still outperform comparing methods like SELDNet, EIN, and SoundDet. Therefore, we can see that larger Transformer-like backbone neural network is important for sound source detection task that requires to jointly identify a sound source's semantic label and predict spatial location. In summary, we can conclude that: 1. both learnable synperiodic filter bank front-end and large Transformer-like backbone help to improve the performance, synperiodic filter bank can be used as a plug-and-play front-end to exiting methods to improve their performance. 2. larger single backbone model is essential for sound source detection, this is echoed by the best-performing method UTSC-Iflytek which uses model ensemble to achieve the best performance.
>
> **Q2**: Not clear how reliable these numbers are.
>
> **A2**: In our experiment, we train each model five times independently and report the averaging performance score. We did not report the standard deviation because many comparing methods did not report it. In our revised version, we report the average standard deviation.
>
> **Q3**: Several related work is missing.
>
> **A3**: We are sorry for missing them. In our revised version, we added the recommended papers and hope it becomes more complete now.
>
> **Q4**: Rewriting the motivations and editing.
>
> **A4**: We really appreciate your patience in reading this paper and kindly pointing out where needs improvement. We followed your guidance and tried best to revise the paper.
>
> 1. we removed "failing" and used "cannot do well" instead.
> 2. we checked the whole paper and change the sentence "filter's length and frequency response are inversely related" to "filter's length and frequency resolution are inversely related".
> 3. other modifications include:
>
> * *old*: "Traditional sound features are human-auditory system biased", *new*: "Traditional sound feature design are motivated influenced by human-auditory system".
> * *old*: Sound source detection has been previously treated as sound structure (Thrun, 2006), sound object (He at al., 2021) and sound event detection and localization (SELD task)", *new*: "Sound source detection has been previously treated as sound structure estimation (Thrun, 2006), sound object detection (He et al., 2021) and sound event detection and localization (SELD) task".
> * *old*: "Our motivation is based the fact that high frequency sound signal can be adequately detected with a shorter filter, enlarging the filter length doesn't necessarily increase the frequency resolution but inevitably reduces the filter's time resolution", *new*: "Our motivation is based on the fact that high-frequency sound signal can be adequately detected with a short filter because its temporal length of one period is small". What we want to emphasize here is that sound signals with different frequencies need filter of different length to be accurately detected.
> * *old*: "The design of this window function is motivated by the mel-scale frequency initialization strategy, in which two neighboring frequencies are more closely spaced in lower frequency part and higher frequency part." *new*: "The design of this window function is motivated by mel-scale frequency initialization strategy: first, the higher frequency of the sound, the narrower of the window size; second, the mathematical relationship between sound frequency and window size is modelled in logarithmic scale."
>  * *old*: "The parameter number is just linear to filter number.", *new*: "The trainable parameter number increases linearly w.r.t. synperiodic filter bank number.".
> *  In Eqn. (5), $l$ means the the final constructed filter length, which is larger than window size. In our implementation, $l$ is the constructed 1D convolution filter length.

---

> > ### Comment · Reviewer_TyRd · 2021-11-27
> > **Thank you for the Revision**
> >
> > I would like to thank the authors for carefully re-editing the manuscript based on my comments and performing the requested ablation study. Given the ablation study showing the most significant part of improvement is indeed coming from the Transformer backbone as several reviewers had pointed out, I am not convinced that the feature extraction re-parameterization is a significant contribution. Additionally the manuscript could still benefit greatly from revision as even in the current state there are a number of ambiguous statements. Although the authors have tried their best to explain why the proposed synperiodic filterbank should be expected to perform better, the whole discussion on time-frequency resolution and uncertainty principle appears a bit superfluous and it is not entirely clear that improving the time-frequency resolution tradeoffs would actually be resulting in performance improvement. It could be that parameterizing the MFCC feature extraction it self could yield improvements which has not been tested.

---

> > > ### Author Response · Authors · 2021-11-30
> > > **More feedback to the new comment**
> > >
> > > We thank the reviewer for the further comments. We also provide more feedback to make our paper clearer.
> > >
> > > **Q1**:  Time-Frequency resolution and uncertainty principle appears a bit superfluous.
> > >
> > > **A1**: The TF resolution and uncertainty principle discussion serves as the motivation of synperiodic filter bank design. They also mark the key difference between synperiodic filter bank and existing methods:
> > >
> > >   1. Since existing methods’s TF resolution map is fixed and single-scaled, Synperiodic filter bank is multi-scale in both time and frequency domain.
> > >   2. Since we cannot achieve optimal resolution in time and frequency domain simultaneously due to uncertainty principle, synperiodic filter bank proposes to learn to optimize the resolution in a data-driven way.
> > >   3. In summary, their discussion helps readers to better understand our motivation and idea novelty. We are thinking about how to improve it.
> > >
> > > **Q2**: Performance gain is mainly due to backbone network.
> > >
> > > **A2**:
> > >   1. Sound sources detection is an extremely challenging task, especially in polyphonic and moving scenario. It thus requires the joint combination of excellent front-end filter bank (e.g. synperiodic filters) and strong backbone network to get promising result, reducing either of them inevitably reduces the performance. Our contribution consists of both the synperiodic filter bank front-end design and Transformer-backbone network, it achieves the best tradeoff between performance and model complexity than all comparing methods.
> > >   2. We specifically point out that UTSC-Iflytek is the best-performing method up to date and it adopts **model ensemble strategy** (e.g. Resnet, inception, etc.), its model size is much larger than our SoundSynp framework. The success of our framework owes to both synperiodic filter bank and backbone network. Removing either of them leads to inferior performance than UTSC-Iflytek.
> > >   3. Return back to the ablation study (Table 2), we highlight the disentanglement experiment between synperiodic filters and backbone network,
> > >       * Using synperiodic filter bank to replace existing methods’ front-end, we observe an average performance gain of 4% (ER), 3% (F), $0.7^\circ$ (LE) and 4% (LR).  We also note that: **The smaller of the backbone network, the more performance gain we get**. For example, SELDNet sees 6% (ER), $1.0^\circ$ (LE) gain when using synperiodic filter bank. It thus shows synperiodic filter bank truly improves the performance dramatically, its prominence is more obvious when the backbone network becomes smaller.
> > >       * In SoundSynp, when we remove synperiodic filter to use traditional features like MFCC and LogMel, we have observed a sharp performance drop: 6% (ER), 5% (F), $2.6^\circ$ (LE) and 8% (LR) on all evaluation metrics, resulting in inferior performance than UTSC-Iflytek. It thus attests the importance of synperiodic filter for improving the performance.
> > >        * We make two tables for clearer understanding. **Top**: performance large gain/drop with/without synperiodic filter bank. **Bottom**: large method performance difference with/without synperiodic filter bank (_synp indicates with synperiodic filter).
> > >
> > > | Method Description      |  Metric Description  |$ER_{20^\circ}$ | $F_{20^\circ}$     |  LE         |  LR       |  Note |
> > > |          :---                       |        :----:               |     :----:             |     :----:           |   :----:     |    :----:   |    :----  |
> > > | Use Synperiodic Filters for Existing Methods (SELDNet)     | Performance Gain      | 4%   | 5% | 0.7 | 7% | The smaller of the backbone, the more improvement brought by Synperiodic filters |
> > > | Remove Synperiodic Filters from SoundSynp     | Performance Drop      | 6%   | 5% | 2.6 | 8% | Sharp performance drop was observed under all metrics |
> > >
> > > | Method      |Model Size |$ER_{20^\circ}$ ($\downarrow$) | $F_{20^\circ}$($\uparrow$)     |  LE ($\downarrow$)        |  LR  ($\uparrow$)    |
> > > |          :---    |      :----:      |      :----:        |     :----:       |   :----:     |    :----:   |
> > > | SELDNet    | 0.5 M        | 0.63 | 0.46 |  23.1 | 0.69 |
> > > | SELDNet_Synp    | 0.5 M        | 0.57 | 0.49 |  22.1 | 0.75 |
> > > |UTSC_Iflytek       |  200 M+    | 0.20 | 0.85 | 6.0    | 0.89 |
> > > |SoundSynp_MFCC | 60 M       | 0.21 | 0.83 | 6.9    | 0.85 |
> > > |SoundSynp_Synp   | 60 M       | 0.16 | 0.88 | 4.3    |  0.93 |
> > >
> > >
> > > **Q3**: lack of MFCC experiment.
> > >
> > > **A3**: Actually, we have contained experiment with MFCC, which is in Table 2. It is also shown in the table above. To summarize, either MFCC, STFT or LogMel extract sound time-frequency in single-scale and non-learnable manner. It is the key difference between them and synperiodic filterbank. Our ablation study of five synperiodic variants in Table 2 further attests this.
> > >
> > > **Q4**: paper needs further revision.
> > >
> > > **A4**: thanks for suggestion, we keep improving it.
> > >
> > > We are not sure if we answered the reviewer's concern, we are happy for more discussion.

---

### Official Review · Reviewer_sfZq · 2021-11-04

**Correctness:** 3
**Technical Novelty And Significance:** 3
**Empirical Novelty And Significance:** 3
**Recommendation:** 6
**Confidence:** 3

**Main Review:**

Accurately estimating sound sources' temporal location can be used in a wide range of machine cognition tasks, such as inference on the type of environment, self-localization, navigation without visual input, smart-home applications, scene visualization systems, and etc.
This manuscript covers a very interesting topic of sound sources' temporal localization.
The authors suggest using Synperiodic Filterbank with the exiting methods to outperform the traditional methods. NOTE, they have proposed a new combination of feature sets and new model structures.
However, I have the following few concerns.
1- The writing of the manuscript requires improvements, there are many ambiguous sections/ paragraphs that require more elaboration. There are many typos and grammatical errors in the manuscript.
2- It is not clear to me whether the improvement in the performance is mainly due to the Synperiodic filterbank or the Synperiodic filterbank+ new structure, which brings the total param size to 60M!
3- the experimental section requires more supporting materials. It is not clear how the training/testing is conducted. It would be more helpful if the authors could elaborate more on some implementation details.
4- The authors in section 3.1 claim their mathematical formulation covers the "moving spatial motion sp" as well. However, they have not discussed/supported this claim in the experimental section.


**Summary Of The Paper:**

I am sorry, by mistake I posted the incomplete review version!

This manuscript proposes a sound source localization method utilizing "synperiodic filter" bank representation as to the input of a deep convolutional neural network.
Authors claim that the convolution of the proposed filterbanks with the raw waveform helps to achieve multi-scale perception in the time domain which results in performance improvement.
The authors demonstrated that their method could outperform some of the state-of-the-art similar work.


**Summary Of The Review:**

This manuscript addresses an ongoing and open problem of sound source localization. I like the main idea in the paper, however, I think this work requires some minor improvements to be in proper shape for this proceedings.

---

> ### Author Response · Authors · 2021-11-15
> **Reply**
>
> **Q1**: The manuscript requires improvement.
>
> **A1**: Thanks for pointing out this. We have carefully revised the whole paper and tried our best to correct the errors/typos and submitted a revised version. In our revised version, we also combined "sound source problem definition" and "sound source clues" section together to make the whole paper more compact and understandable.
>
> **Q2**: Not clear if the improvement is mainly due to Synperiodic filter bank alone or Synperiodic filter bank +  new structure.
>
> **A2**: In our first version submission, we have conducted two ablation studies (shown in Table 2)
>
> 1. We replace SELDNet, EIN and SoundDet front-end head with our proposed synperiodic filter bank. Experimental result shows that introducing synperiodic filter bank to existing approaches helps to improve their corresponding performance. It thus shows our proposed synperiodic filter bank can be used as a plug-and-play front-end by existing approaches to improve their performance on sound source detection task, without introducing explicit extra parameters and computational burden.
>
> 2. Moreover, we test five variants of synperiodic filter bank. For example, we find out that reducing the multi-scale perception strategy of synperiodic filter bank to single-scale perception in either time domain or frequency domain reduces the performance. This internal synperiodic filter bank variants ablation study shows the necessity of each part of synperiodic filter bank design.
>
> To disentangle the synperiodic filter bank front-end and Transformer-like backbone, in our revised paper, we have added another ablation study (also given in Table 2), in which we replace our proposed SoundSynp's synperiodic filter bank with traditional hand-engineered features including MFCC and LogMel. The corresponding experimental result shows that using traditional sound features leads to inferior performance. In sum, we can conclude from the three ablation studies that:
>
> *  The performance improvement is due to the joint contribution of synperiodic filter bank and Transformer-like backbone network structure.
>
> *  Removing either Transformer-like backbone network (ablation study 1) or synperiodic filter bank (ablation study 2) inevitably reduces the performance.
>
> * Transformer-like backbone network and synperiodic filter bank can be disentangled to solve other sound related tasks separately.
>
> **Q3**: More training detail and implementation detail are needed.
>
> **A3**: In our first version, we provided detailed network architecture and training detail in the Appendix section (A.2 and Table 4). In our revised version, we move training details to the main paper (Experiment section, page 8) for direct reading.
>
> **Q4**: Moving spatial motion $sp$ discussion.
>
> **A4**: Sound source can be either stationary or moving (i.e. footstep). The DCASE dataset contains both stationary and moving motion sound sources. Existing methods including SELDNet, EIN, UTSC-Iflytek and our SoundSynp formulate it as a frame-wise sound source semantic label classification and  spatial location regression task. It means that, for moving spatial motion sound source, two neighboring frames' predicted spatial locations should be different (similarly, if the sound source is stationary, the neighboring spatial location predictions should be the same). Therefore, in segment-based evaluation metric, a correctly detected sound source not only has the right semantic label, but also spatially lie close enough to ground truth spatial location (see [link](http://dcase.community/challenge2020/task-sound-event-localization-and-detection) for detailed discussion). In event-based evaluation metric, the spatial closeness and semantic label correctness are taken into consideration at the same time to compute the mAP and mAR score. In summary, current sound source detection evaluation unifies stationary and moving sound source motion so that they can be evaluated within the same metric. Moreover, the qualitative representation in Fig. 3 shows a moving spatial motion sound sources (as its azimuth location varies along the time axis), we can see that SoundSynp produces more consistent and complete sound source prediction than all other comparing methods.

---

### Author Response · Authors · 2021-11-15
**Rebuttal Summary**

We sincerely thank all reviewers for reviewing our paper. According to their comments, we have submitted a revised version of our paper that contains three main revisions:

1.  Word writing, typo/error correction. Specifically, we tried our best to minimise the ambiguous word using, unclear writing, typo/errors. We also rewrite some sentences to make our idea more clear and understandable.

2. Paper re-organization. To make the whole paper more compact and smooth to read, we re-organized the paper from two three aspects:
    * First, we combined ''sound source problem definition'' and ''sound source clues'' together to make the whole paper more compact.
    * Second, we rewrite Eqn. (5) and add an extra equation to explicitly illustrate the relationship between each synperiodic filter's center frequency $w_i$, window size $\sigma_i$ and periodicity term $\rho$.
    * Third, we re-organize the experiment section. 1.  split direction-of-arrival estimation, physical location estimation and conclusion with subsection title. 2. give more detailed discussion on experiment dataset, evaluation metrics. 3. at the end of experiment section, we explicitly give a "Discussion and Conclusion" subsection to summarize the paper and highlight the novelty and contribution of our work.

3. We added another group of ablation study: replacing our proposed SoundSynp neural network's synperiodic filter bank with traditional fixed sound features-MFCC and LogMel. We observed performance drop (see Table 2 in our revised version), but the performance is still better than comparing methods.  In our first submission, we have conducted two ablation studies: first, replacing comparing methods (SELDNet, EIN, SoundDet) head with our proposed synperiodic filter bank - performance improves. second, we tested five synperiodic filter bank construction variants - performance drops. In sum, with all the three different ablation studies, we can conclude that:
     * Our proposed synperiodic filter bank can be used as a plug-and-play front-end for sound feature extractor by other methods to improve their performance, without explicitly introducing extra parameters and computational burden.
     * Our proposed SoundSynp's best performance is due to both the synperiodic filter bank and the large Transformer-like backbone. Removing any of them will inevitably reduce the performance.  Since sound source detection is a challenging task due to the polyphonicity characteristic and large scale-variance problem, designing a large model is essential to obtain good performance. This is echoed by UTSC-Iflytek method which adopted model ensemble to achieve best performance in the leaderboard (see [leaderboard link](http://dcase.community/challenge2020/task-sound-event-localization-and-detection#results)), it merges various large models to improve the performance.
     * Our proposed SoundSynp's learnable synperiodic filter bank front-end and Transformer-like backbone can be easily disentangled to solve other sound related tasks separately.

We also want to emphasize here that we test the efficiency and excellence of our framework on sound source's physical location (directly estimating sound source $[x, y, z]$ coordinate) estimation task as well. This is in contrast with all comparing methods that simply do experiments on  direction-of-arrival (DoA ) task.

---

### Decision · Program_Chairs · 2022-01-20

**Decision:**

Reject

**Comment:**

This work studies the task of sound source localization from multi-channel audio. An approach to design a wavelet-like filter bank for audio feature extraction is proposed.

After discussion, all reviewers have given this work borderline ratings. Concerns were raised about the quality of the writing, missing related work, experimental methodology, and especially with regards to confounding factors in the experimental results, which make it difficult to assess the individual merit of the different components in the proposed system (i.e. features vs. transformer model). The authors have addressed this to some extent with additional experiments in the updated version of the manuscript, but this revealed that the gains obtained by the proposed feature extraction method in isolation are actually quite modest. This is in contrast with how the paper is written, with much more emphasis on this particular contribution than seems to be warranted by the empirical results.

Additionally, I believe the manuscript would benefit from a more careful and thorough revision to improve clarity and accessibility, beyond what is possible within a single review cycle. Therefore I am recommending rejection.